# Aberrant regulation of a poison exon caused by a non-coding variant in a mouse model of *Scn1a*-associated epileptic encephalopathy

Yuliya Voskobiynyk[1], Gopal Battu[2], Stephanie A. Felker[2,3], J. Nicholas Cochran[2], Megan P. Newton[4], Laura J. Lambert[4], Robert A. Kesterson[4], Richard M. Myers[2], Gregory M. Cooper[2], Erik D. Roberson[1]*, Gregory S. Barsh[2]*

**1** Center for Neurodegeneration and Experimental Therapeutics, Alzheimer's Disease Center, and Evelyn F. McKnight Brain Institute, Departments, of Neurology and Neurobiology, University of Alabama at Birmingham, Birmingham, AL, United States of America, **2** HudsonAlpha Institute for Biotechnology, Huntsville, AL, United States of America, **3** Department of Department of Biotechnology Science and Engineering, University of Alabama in Huntsville, Hunstville, AL, United States AL, United States of America, **4** Department of Genetics, University of Alabama at Birmingham, Birmingham, AL, United States of America

☯ These authors contributed equally to this work.
* gbarsh@hudsonalpha.org (GSB); eroberson@uabmc.edu (EDR)

**Data Availability Statement:** RNA-seq data is available at the NCBI Short Read Archive

## Abstract

Dravet syndrome (DS) is a developmental and epileptic encephalopathy that results from mutations in the Na$_v$1.1 sodium channel encoded by *SCN1A*. Most known DS-causing mutations are in coding regions of *SCN1A*, but we recently identified several disease-associated *SCN1A* mutations in intron 20 that are within or near to a cryptic and evolutionarily conserved "poison" exon, 20N, whose inclusion is predicted to lead to transcript degradation. However, it is not clear how these intron 20 variants alter *SCN1A* expression or DS pathophysiology in an organismal context, nor is it clear how exon 20N is regulated in a tissue-specific and developmental context. We address those questions here by generating an animal model of our index case, NM_006920.4(SCN1A):c.3969+2451G>C, using gene editing to create the orthologous mutation in laboratory mice. *Scn1a* heterozygous knock-in (+/*KI*) mice exhibited an ~50% reduction in brain *Scn1a* mRNA and Na$_v$1.1 protein levels, together with characteristics observed in other DS mouse models, including premature mortality, seizures, and hyperactivity. In brain tissue from adult *Scn1a* +/+ animals, quantitative RT-PCR assays indicated that ~1% of *Scn1a* mRNA included exon 20N, while brain tissue from *Scn1a* +/*KI* mice exhibited an ~5-fold increase in the extent of exon 20N inclusion. We investigated the extent of exon 20N inclusion in brain during normal fetal development in RNA-seq data and discovered that levels of inclusion were ~70% at E14.5, declining progressively to ~10% postnatally. A similar pattern exists for the homologous sodium channel Na$_v$1.6, encoded by *Scn8a*. For both genes, there is an inverse relationship between the level of functional transcript and the extent of poison exon inclusion. Taken together, our findings suggest that poison exon usage by *Scn1a* and *Scn8a* is a strategy to regulate channel expression during normal brain development, and that mutations recapitulating a fetal-like pattern of splicing cause reduced channel expression and epileptic encephalopathy.

(SRP269180), and at the NCBI Gene Expression Omnibus (GSE153461).

**Funding:** The work was funded by grants from the National Institutes of Health to GMC, GSB, and EDR. Services obtained from the UAB Transgenic & Genetically Engineered Model Systems Core Facility (RAK) are supported by grants from the National Institutes of Health (P30 CA13148, P30 AR048311, P30 DK074038, P30 DK05336, P60 DK079626, U01HG007301, and R01MH110472). The funders had no role in study design, data collection and analysis, decision to publish, or preparation of the manuscript.

**Competing interests:** The authors have declared that no competing interests exist.

## Author summary

Dravet syndrome (DS) is a neurological disorder affecting approximately 1:15,700 Americans that causes generalized epilepsy and associated complications. While most patients have a mutation in the *SCN1A* gene that encodes the Na$_v$1.1 voltage-gated sodium channel, about 20% do not have a mutation identified by exome or targeted sequencing. Recently, we identified variants in intron 20, a noncoding region of *SCN1A*, in some DS patients. We hypothesized that these variants alter *SCN1A* transcript processing, decrease Na$_v$1.1 function, and lead to DS pathophysiology via inclusion of exon 20N, a "poison" exon that leads to a premature stop codon. In this study, we generated a knock-in mouse model, *Scn1a+/KI*, of one of these variants, which resides in a genomic region that is extremely conserved across vertebrate species. We found that *Scn1a+/KI* mice have reduced levels of *Scn1a* transcript and Na$_v$1.1 protein and develop DS-related phenotypes. We find that transcripts from brains of *Scn1a+/KI* mice show elevated rates of *Scn1a* exon 20N inclusion. We also explored the relationship between exon 20N inclusion and *Scn1a* expression during development, and found that, during brain development when *Scn1a* expression is low, exon 20N inclusion is high; postnatally, as *Scn1a* expression increases, there is a corresponding decrease in exon 20N usage. Expression of another voltage-gated sodium channel transcript, *Scn8a* (Na$_v$1.6), was similarly regulated. Together, these data demonstrate that poison exon inclusion is a conserved mechanism to control sodium channel expression in the brain, and that an intronic mutation that disrupts the normal developmental regulation of poison exon inclusion leads to reduced Na$_v$1.1 and generalized epilepsy.

## Introduction

Dravet syndrome (DS) is a developmental and epileptic encephalopathy (DEE) characterized by intractable seizures, developmental delay, speech impairment, ataxia, hypotonia, sleep disturbances, and other health problems [1]. In the U.S., DS incidence is 1 per 15,700 [2], and 73% of patients die before the age of 10 years [3].

The most frequent cause of DS are loss-of-function mutations of *SCN1A*, which encodes the type I voltage-gated sodium channel (Na$_v$1.1) alpha subunit, part of a larger family of nine sodium channel proteins (Na$_v$1.1 –Na$_v$1.9) that control neuronal excitability [4–8]. Pathogenic *SCN1A* mutations are generally heterozygous and often occur *de novo* in DS. DS-associated *SCN1A* mutations lead to a loss of Na$_v$1.1, which is predominantly expressed in inhibitory GABAergic interneurons, so loss of function leads to network disinhibition [5–9]. Importantly, the molecular mechanisms for Na$_v$1.1 loss of function differ between various *SCN1A* mutations; many cause nonsense-mediated RNA decay, while other missense mutations affect Na$_v$1.1 stability or function [4].

Only 80% of DS patients have pathogenic *SCN1A* variants detectable within coding exons [10], suggesting that variants in noncoding regions near *SCN1A* may contribute to disease in some patients. A genomic analysis of 640 DEE patients found that five patients harbored rare variants predicted to be deleterious within a highly conserved region deep within *SCN1A* intron 20 [11]. A 64-bp segment within this region can be alternatively spliced and included as an exon termed 20N [11]. Exon 20N is known as a poison exon because it is predicted to lead to a truncated *SCN1A* isoform due to a stop codon that arises with the frameshift caused by the 64-bp inclusion [12]. Several of the intron 20 variants identified in DEE patients increased

inclusion of poison exon 20N in splice reporter assays in non-neuronal cells [11]. As a result, we hypothesized that variant-induced aberrant inclusion of *SCN1A* poison exon 20N is a pathophysiologic mechanism for Na$_v$1.1 loss of function in DEE patients [11].

Our prior work on non-coding variation and poison exon inclusion in DS was carried out in non-neuronal cultured cells with artificial constructs and did not determine if any of the non-coding variants could recapitulate the phenotype of DS in an organismal context. Here, we report the construction and analysis of a mouse model for a *SCN1A* variant, NM_006920.4 (SCN1A):c.3969+2451G>C (hereafter, c.3969+2451G>C), that we identified in our index patient and that lies within the alternatively spliced poison exon, 20N [11]. Our results provide rigorous evidence of causality for a non-coding variant, allow direct measurement of poison exon usage *in vivo*, and give new insight into the normal function of poison exons for sodium channel genes and the consequent relationship to human genetic disease.

## Results

### Evolutionary conservation in intronic regions harboring a DS-causing variant

The index patient with an *SCN1A* c.3969+2451G>C variant was an 11-year old male originally diagnosed with febrile seizures plus due to febrile generalized tonic clonic seizures at 23 months and afebrile GTCS at 26 months. Development perinatally and prior to seizure onset was reported as normal, as was a head CT at 17 months and an MRI at 2 years of age. Mild speech delay resolved by 11 years of age.

We first examined evolutionary conservation in the region surrounding the variant as a prerequisite to identifying the orthologous variant in mice. Human intron 20 is ~8 kb, within which there exist three highly conserved segments of several hundred nucleotides in length (**Fig 1A**). Exon 20N and the surrounding region is highly conserved as indicated by quantitative assessment with genomic evolutionary rate profiling (GERP) (**Fig 1B**) and alignment across 77 vertebrates (**S1 Fig.**). The G>C substitution in our index patient lies within exon 20N and is perfectly conserved along with neighboring nucleotides in the mouse. *SCN1A* transcripts that contain this exon 20N are "poisoned" due to a frameshift and consequent premature termination codon in exon 21 (**Fig 1C and 1D**); the same is true for mouse *Scn1a*.

### *Scn1a* mRNA and protein levels are reduced in the brains of *Scn1a +/KI* mice

We used CRISPR/Cas9 gene editing [13] to generate the mouse mutation NC_000068.7: g.66293870C>G (GRCm38.p6), orthologous to the *de novo* mutation in our index patient, c.3969+2451G>C. (In both humans and mice, the gene lies on the minus strand; references to the human G>C *de novo* variant are based on the transcript, NC_000068.7, as in **Fig 1**, while references to the mouse gene-edited C>G variant are based on genomic coordinates). A guide RNA located upstream of the variant position (**Fig 1B**) was used together with a template for homology-directed repair, and the ribonucleoprotein complex was microinjected into C67BL/6J zygotes as described in Materials and methods. Founder animals carrying the C>G variant were backcrossed to C57BL/6J mice, and all genotypes were determined by Sanger sequencing. Animals carrying one allele of the edited variant are termed *Scn1a +/KI* and compared to non-mutant *Scn1a +/+* littermates.

In brain tissue from postnatal *Scn1a +/KI* mice, qRT-PCR for an amplicon between exons 19 and 20 (Materials and methods) indicated an ~50% reduction in levels of *Scn1a* mRNA (**Fig 2B**). Analysis of RNA-seq data from +/+ and +/KI brains yielded a similar result (**Fig 2C**).

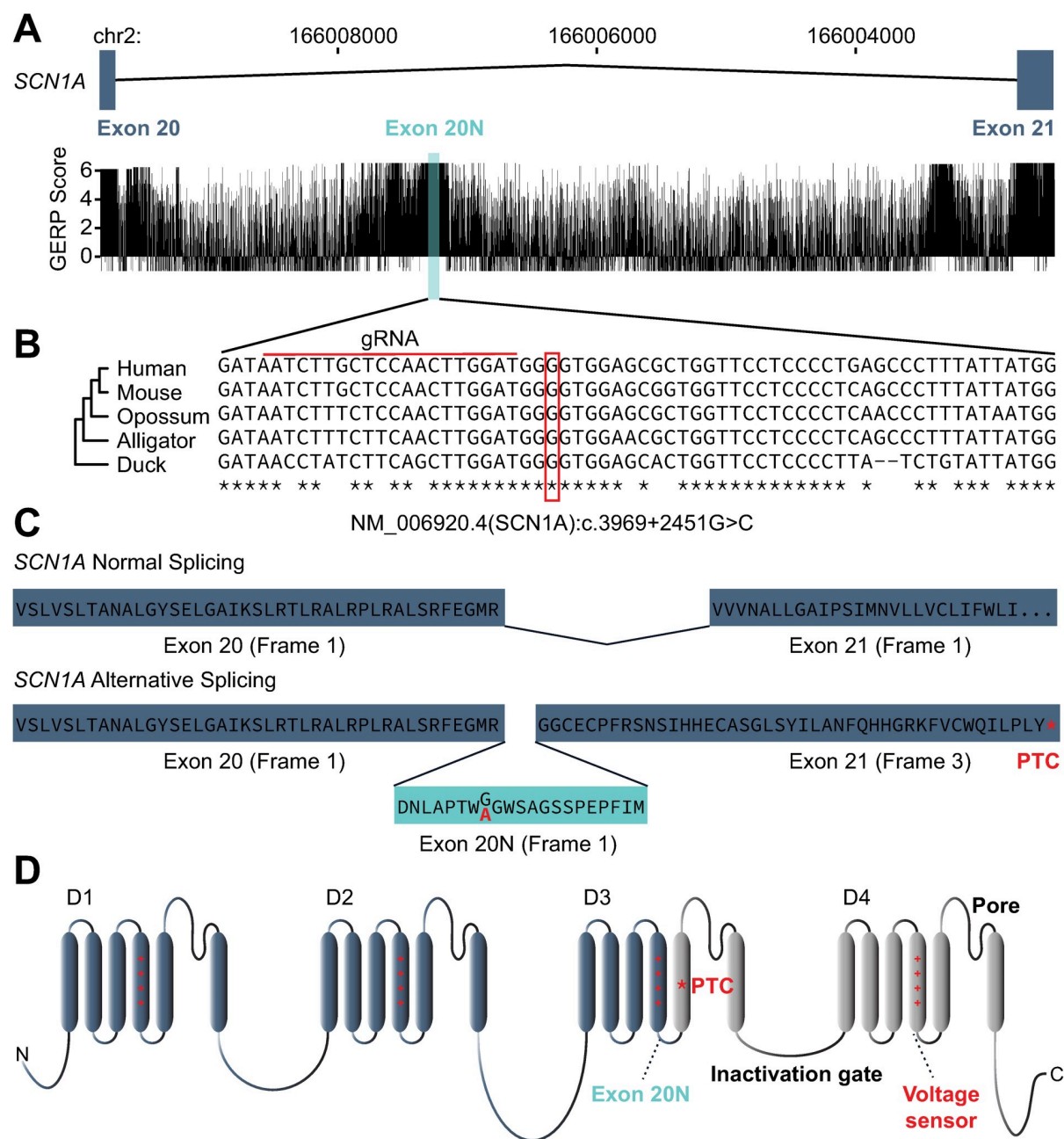

**Fig 1. The non-coding Dravet Syndrome–causing variant, NM_006920.4(SCN1A):c.3969+2451G>C, is present in a highly conserved region.** (A) The alternate exon 20N (shaded rectangle) is highly conserved, with GERP scores that are comparable to canonical exons in *SCN1A*. (B) Multiple alignment in the 64bp *SCN1A* 20N region of human, mouse, opossum, alligator, and duck, modified from the Multiz Alignment of 100 Vertebrates track from the UCSC Genome Browser (Fig 1). The red box indicates the position of the variant NM_006920.4 (SCN1A):c.3969+2451G>C in our index patient, orthologous to NC_000068.7:g.66293870C>G (GRCm38.p6) in the mouse genome. The red line indicates the position of the guide RNA used for CRISPR/Cas9 gene editing. (C) Alternative splicing of intron 20 in *SCN1A*. Inclusion of exon 20N (bottom) results in a frame shift and hence a premature termination codon (PTC) in exon 21. The c.3969+2451G>C variant also results in a Gly-Ala (red) substitution within exon 20N. (D) Exon 20N would be in the intracellular loop connecting the fourth and fifth transmembrane voltage sensing regions of the third SCN1A homologous domain (D3) but brings a premature termination codon (PTC) in frame resulting in nonsense-mediated RNA decay.

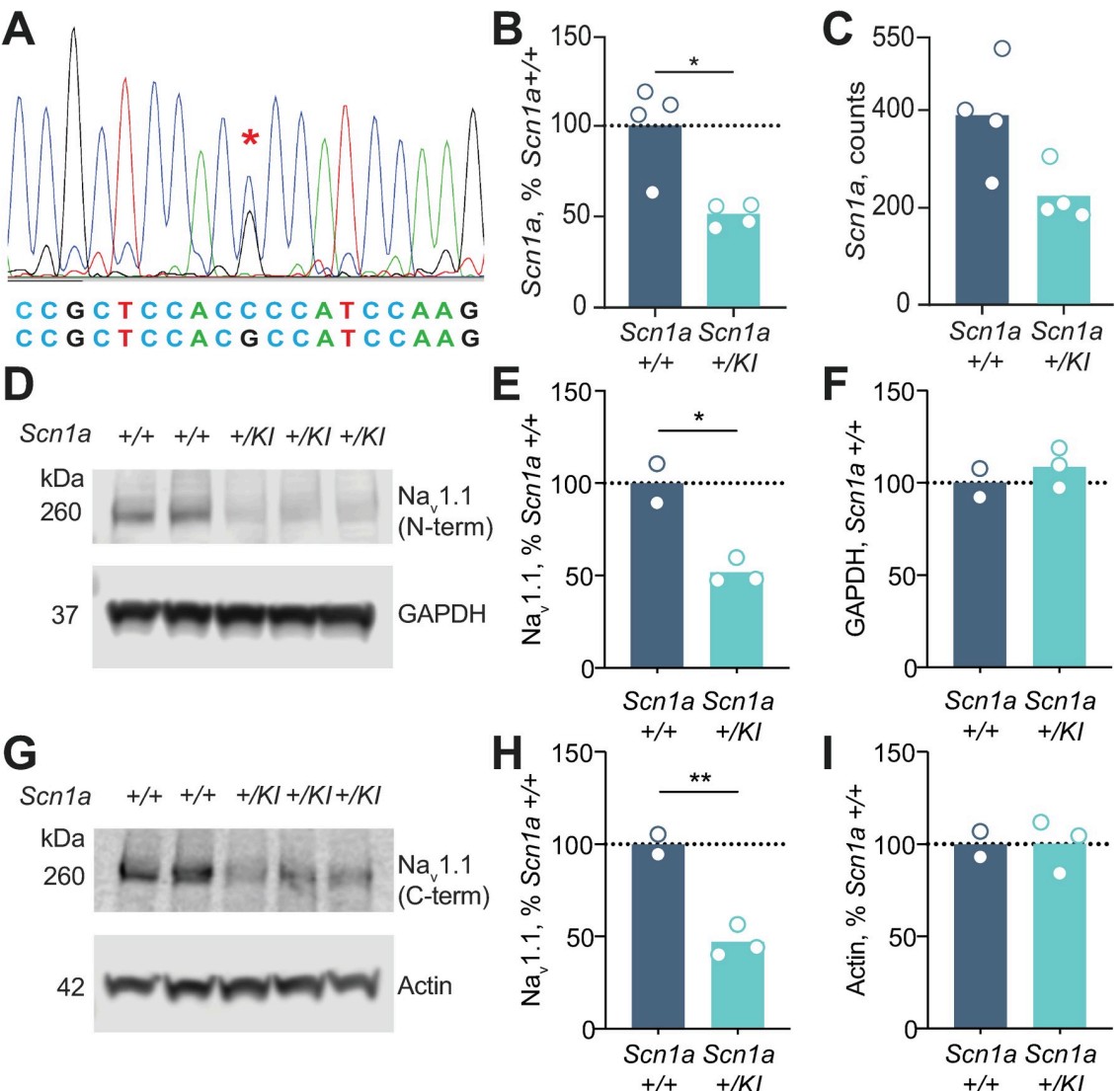

**Fig 2. *Scn1a* mRNA and Na$_v$1.1 protein levels are reduced in *Scn1a+/KI* mice.** (A) Sanger sequence confirmation of *Scn1a+/KI* mouse with the gene-edited intron 20 C>G variant. (B) Brain mRNA levels in *Scn1a+/+* and *Scn1a+/KI* mice using qRT-PCR. Relative expression of *Scn1a* vs. the control gene *Tbp* ($n$ = 4–4, 11.64 ± 2.90 months, Student's unpaired t-test, $p$ = 0.0251). *Scn1a+/KI* mice have ~50% less *Scn1a* mRNA than *Scn1a +/+* mice. (C) RNA-seq counts (normalized to sequencing library size by DEseq2) of *Scn1a* mRNA in whole brains of *Scn1a+/+* and *Scn1a+/KI* mice (n = 4–4, 11.64 ± 2.90 months, Student's unpaired t-test, $p$ = 0.0541). *Scn1a+/KI* mice have about 42% less *Scn1a* mRNA than *Scn1a+/+* mice. (D) Levels of Na$_v$1.1, the sodium channel encoded by *Scn1a*, are reduced in frontal cortex of *Scn1a+/KI* vs. *Scn1a+/+* mice using rabbit anti-Na$_v$1.1 antibody from Alomone Labs, which recognizes an N-terminal epitope. GAPDH served as a loading control. (E) Quantification of Na$_v$1.1 levels from the blot in D ($n$ = 2–3, 17.7 ± 0.96 months, Student's unpaired t-test, $p$ = 0.0142). (F) Quantification GAPDH protein levels from the blot in D ($n$ = 2–3, 17.7 ± 0.96 months, Student's unpaired t-test, $p$ = 0.4459). (G) Na$_v$1.1 levels using anti-Na$_v$1.1 Antibodies Incorporated antibody, which recognizes a C-terminal epitope. Actin served as a loading control. (H) Quantification of Na$_v$1.1 protein levels from the blot in G (n = 2–3, 17.7 ± 0.96 months, Student's unpaired t-test, $p$ = 0.0059). (I) Quantification of actin protein levels from the blot in G ($n$ = 2–3, 17.7 ± 0.96 months, Student's unpaired t-test, $p$ = 0.9859, respectively). $^*p < 0.05$ and $^{**}p < 0.01$.

We assessed Na$_v$1.1 protein levels with antisera targeting C-terminal or N-terminal epitopes; in both cases, levels of full-length protein (260 kDa) was reduced by ~50% (**Fig 2D–2I**), and there was no evidence of a truncated protein (157 kDa) that would otherwise correspond to the protein predicted from a transcript that contains exon 20N (**S2A Fig**). Taken together, these results indicate that the variant we introduced into *Scn1a* leads to the absence of Na$_v$1.1,

likely due to nonsense-mediated decay of a transcript that contains exon 20N and a downstream premature termination codon.

## *Scn1a +/KI* mice exhibit Dravet syndrome–like phenotypes

In crosses between *Scn1a +/+* and *+/KI* mice, litter sizes appeared normal, but spontaneous seizures were occasionally observed (video in **Supplementary File 1**), and there was a significant reduction in the expected 50% proportion of *+/KI* mice after weaning from *+/+* x *+/KI* matings (20 *+/KI* vs. 53 *+/+*; $p = 0.004$). Surviving *Scn1a +/KI* mice exhibited ~40% mortality at 3 months and ~55% mortality by 18 months (**Fig 3A**). These observations are consistent with previous DS mouse models on the C57BL/6J background or a mixed C57BL/6 and 129 background, in which the onset of seizures in heterozygotes begins around P21, with ~50% mortality by 6 months [7, 8].

We tested *Scn1a +/KI* mice that survived to adulthood in a battery of behavior assays [14–17] to investigate if they developed behavioral deficits reported in other DS mouse models [18,

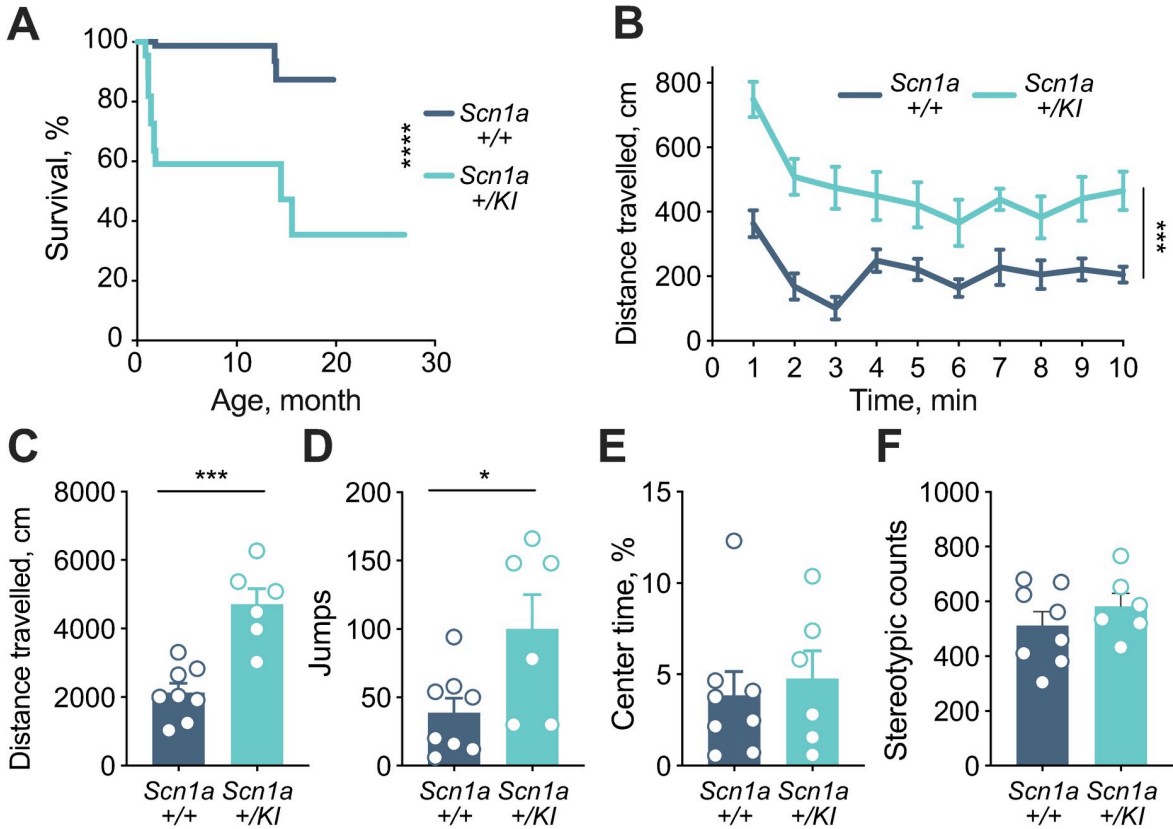

**Fig 3. *Scn1a+/KI* mice exhibit premature mortality and a hyperactivity phenotype.** (A) Kaplan-Meier analysis showed severe premature mortality in *Scn1a+/KI* mice ($n = 22$–93, Log-rank (Mantel-Cox) test, $p < 0.0001$). This analysis started at weaning, when genotyping was performed, and so does not include additional mortality observed in litters prior to weaning. (B) Distance travelled in the open field as a function of time is increased in *Scn1a+/KI* mice (n = 6–8, 13.81 ± 0.47 months, two-way RM-ANOVA, interaction $p = 0.0150$, main effects of time $p < 0.0001$ and genotype $p = 0.0003$). (C) Total distance travelled during 10 minutes in the open field is higher in *Scn1a+/KI* mice ($n = 6$–8, 13.81 ± 0.47 months, Student's unpaired t-test, $p = 0.0003$). (D) Vertical jumps in the open field apparatus are higher in *Scn1a+/KI* mice ($n = 6$–8, 13.81 ± 0.47 months, Student's unpaired t-test, $p = 0.0305$). (F) No difference between genotypes were found in stereotypic counts in the open field ($n = 6$–8, 13.81 ± 0.47 months, Student's t-test, $p = 0.3444$). (E) No difference between genotypes in percent time spent in the center of the open field ($n = 6$–8, 13.81 ± 0.47 months, Student's unpaired t-test, $p = 0.6598$). All data are expressed as mean +/–SEM; $^*p < 0.05$, $^{***}p < 0.001$ and $^{****}p < 0.0001$.

19]. Consistent with phenotypes of other DS models [14, 20], *Scn1a +/KI* mice exhibited hyperactivity in the open field (**Fig 3B–3F**), including increased distance travelled (**Fig 3B and 3C**) and jumps (**Fig 3D**). Notably, percent time in the center of the open field and stereotypic counts were similar in *Scn1a +/KI* mice compared to *Scn1a +/+* mice (**Fig 3E and 3F**), indicating no apparent evidence of increased anxiety. In addition, testing in an elevated plus maze, another behavioral assay for anxiety-related phenotypes, [16, 17, 21], revealed no differences in the time spent in open or closed arms, nor total entries into the open and closed arms of the maze (**S3A–S3C Fig**). In a Y maze assay [16, 17, 22], no differences in spontaneous alternations were observed between *Scn1a +/+* and *+/KI* mice (**S3D and S3E Fig**), indicating no apparent deficits in short-term memory. Lastly, *Scn1a +/KI* mice did not show any behavioral deficits in the tube test of social dominance [23] (**S4A Fig.**) or three-chamber sociability test [24] (**S4B–S4D Fig.**). Overall, our results on DS-associated behavioral phenotypes in *Scn1a +/KI* are similar to what has been reported previously in other DS mouse models. Taken together with the molecular characterization (**Fig 2**), these results demonstrate that the mouse models the molecular pathophysiology of a conserved non-coding mutation in exon 20N and provides compelling evidence of its pathogenicity in DS.

## Retention of exon 20N in *Scn1a +/KI* mice

We designed several qPCR primer sets to detect mRNA transcripts either containing or excluding exon 20N (**Fig 4A**), after reverse transcription with random primers (Materials and methods). Amplicon 1 spans from exon 20 to exon 21 and generates a 56-bp product without exon 20N or a 120-bp product when exon 20N is included. In brain RNA from animals aged 1.9 mo–19 months, levels of the larger transcript reflecting exon 20N inclusion were undetectable in *Scn1a +/+* mice, but easily detectable in *Scn1a +/KI* mice (**Fig 4B** and **4C**). A second set of primers spans the introns between exons 20 and 20N, and between exons 20N and 21, allowing measurement of exon 20N-containing transcripts as a 96-bp product, amplicon 2, that can be directly compared to a 111-bp product, amplicon 3, that spans exons 19 and 20 (**Fig 4A**).

Expressed as a percentage of amplicon 2/amplicon 3, exon 20N is included in 0.97% of *Scn1a* transcripts in *+/+* mice, and in 4.8% of *Scn1a* transcripts in *+/KI* mice (**Fig 4D**). Similar results were obtained after reverse transcription with oligodT. Assuming an additive model in which the presence of the *KI* allele does not influence activity of the + allele, and vice versa, we conclude that gene-edited variant leads to a ~9 to 10-fold increase in stable *Scn1a* transcripts that contain exon 20N. However, it is important to note that the levels of normal *Scn1a* mRNA and protein are reduced ~50% in *+/KI* compared to *+/+* mice (**Fig 2**). Assuming that the *KI* variant does not affect transcriptional initiation of *Scn1a* or transcriptome composition (below), this implies that nearly all transcription from the *KI* allele contains exon 20N, most of which is degraded and does not give rise to a functional protein.

As an alternative approach to evaluating usage of exon 20N, we constructed and analyzed RNA-seq libraries from brain tissue of four *Scn1a +/KI* mice and four *Scn1a+/+* littermates. The number of reads that aligned to all *Scn1a* exons was 387.83 ± 56.49 and 223.24 ± 27.58 in *+/+* and *+/KI* mice, respectively (**Fig 2C**), but the number of reads that aligned to exon 20N were (0,0,1,0) and (0,0,0,0) in *+/+* and *+/KI* mice, indicating that most transcripts that contain exon 20N are degraded.

## Potential function of poison exons in *Scn1a* and *Scn8a*

Aberrant regulation of poison exons as a pathogenetic mechanism, and conservation of that mechanism in humans and mice, raises the more general question of how and why *Scn1a* poison exons are used normally during development and differentiation. Voltage-gated sodium

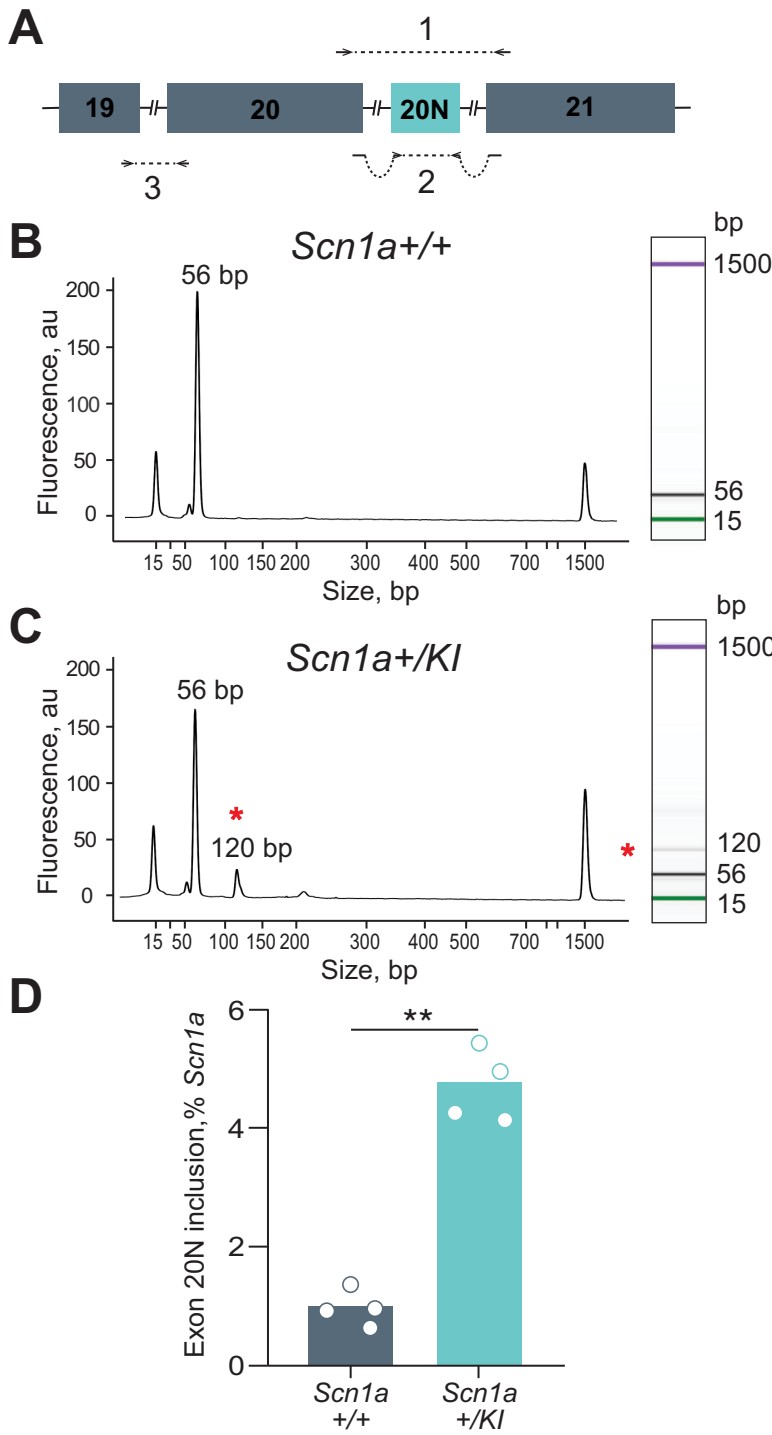

**Fig 4. Increased inclusion of Exon 20N in *Scn1a+/KI* brains.** (A) The positions of qPCR amplicons to quantify *Scn1a* mRNA transcripts. Amplicon 1 detects two isoforms (56bp and 120bp) of the *Scn1a* transcript, with the longer isoform reflecting exon 20N inclusion. Amplicon 2 quantifies only the Exon 20N-containing transcript. Amplicon 3 quantifies the total *Scn1a* mRNA levels including the transcript with Exon 20N. (B) Bioanalyzer evaluation of RNA from *Scn1a +/+* mouse brain amplified with amplicon 1, showing a single *Scn1a* peak at 56 bp. The peaks at 15-bp and 1500-bp are size markers recommended and supplied by the manufacturer. (**C**) Bioanalyzer evaluation of RNA from *Scn1a+/KI* mouse brain amplified with amplicon 1, showing a second peak at 120 bp, representing inclusion of exon 20N. The 120-bp amplicon containing the 64-bp exon 20N is denoted with a red asterisk. (D) *Scn1a+/KI* mice had increased levels of the exon 20N–containing *Scn1a* transcript, measured using amplicon 2. The levels of exon 20N transcript

(amplicon 2) expressed as a percentage of the total *Scn1a* levels (amplicon 3) using the formula (amplicon 2 levels)/(amplicon 3 levels)*100. (n = 4, 11.64 ± 2.90 months, Student's unpaired t-test, *p* = 2e-4). **$p < 0.01$.

channels (Na$_v$1.1–Na$_v$1.9) have distinct developmental and regional patterns of expression [25–28]. Na$_v$1.1, the alpha subunit encoded by *Scn1a*, rises after a lag phase to adult levels during the second to fourth postnatal weeks in both mouse and rat brains, and is expressed primarily in a subset of GABAergic interneurons [25–27]. The homologous protein Na$_v$1.6, encoded by *Scn8a*, has a similar pattern of developmental expression to Na$_v$1.1, but is expressed primarily in glutamatergic neurons. Previous work from Meisler and colleagues on *Scn8a* has identified a poison exon, 18N, whose expression is highest in fetal brain [29]. Inclusion of exon 18N in *Scn8a* is regulated by several RNA binding proteins [30, 31], and for which a "fail-safe" mechanism has been proposed to prevent the synthesis of active protein in cells or tissues where it would be deleterious. We explored that idea for *Scn1a* by first measuring expression of alternative isoforms in different tissues of +/+ and +/*KI* mice using isoform-specific amplicons as shown in Fig 4A. In heart, kidney, liver, and lung of +/+ and +/*KI* mice, *Scn1a*, with or without exon 20N, was expressed at very low levels as detected by qRT-PCR (**Table 1**).

We did not have access to fetal tissues from *Scn1a* +/*KI* mice, but we analyzed usage of *Scn1a* exon 20N in non-mutant mice by analyzing a previously generated RNA-seq dataset of mouse cortex at multiple developmental timepoints [32]. Expressed as a proportion of reads that align to exon 20N compared to all other exons, ~70% of *Scn1a* transcripts include 20N at E14.5, gradually decreasing to <10% by P30, and remaining minimal throughout adult life (**Fig 5A**). This pattern is inversely correlated with the overall level of *Scn1a* mRNA, inferred from the total number of reads (**Fig 5A**). Thus, as exon 20N usage decreased, more *Scn1a* mRNA was produced, consistent with the poison exon inclusion being used to reduce *Scn1a* levels during development. We used the same dataset to evaluate usage of poison exon 18N in *Scn8a* and observed a very similar pattern (**Fig 5B**). This confirms the results of Meisler and colleagues [29], and suggests that poison exons for both sodium channel genes serve a similar function.

## Discussion

Here, we generated a knock-in mouse model of an intronic variant, previously identified as a *de novo* mutation in a patient with DS, to explore its effects on *Scn1a* expression and function

**Table 1.** *Scn1a* expression in different tissues[a].

| Tissue | Genotype | *Scn1a* Ct | *Tbp* Ct | △Ct | 20N Ct |
|---|---|---|---|---|---|
| **Lung** | *Scn1a* +/+ | 35.6 | 22.7 | 12.9 | UD[b] |
| | *Scn1a* +/*KI* | 33 | 22.8 | 10.2 | 36.8 |
| **Liver** | *Scn1a* +/+ | 36.9 | 23 | 13.9 | UD[b] |
| | *Scn1a* +/*KI* | UD[b] | 22.4 | - | UD[b] |
| **Kidney** | *Scn1a* +/+ | 30.5 | 21.5 | 9 | 35.8 |
| | *Scn1a* +/*KI* | 31.1 | 21.8 | 9.3 | 35.2 |
| **Heart** | *Scn1a* +/+ | 36.1 | 25.4 | 10.7 | UD[b] |
| | *Scn1a* +/*KI* | 34.6 | 24.9 | 9.7 | 38 |
| **Brain** | *Scn1a* +/+ | 21.6 | 23.7 | -2.1 | 28.3 |
| | *Scn1a* +/*KI* | 22.5 | 23.8 | -1.3 | 27 |

a Levels of *Scn1a* mRNA and exon 20N as measured by quantitative RT-PCR-based Ct values using amplicons 3 and 2, respectively (Fig 4A), compared to the housekeeping control gene *Tbp*. UD = Undetectable.
b Undetectable

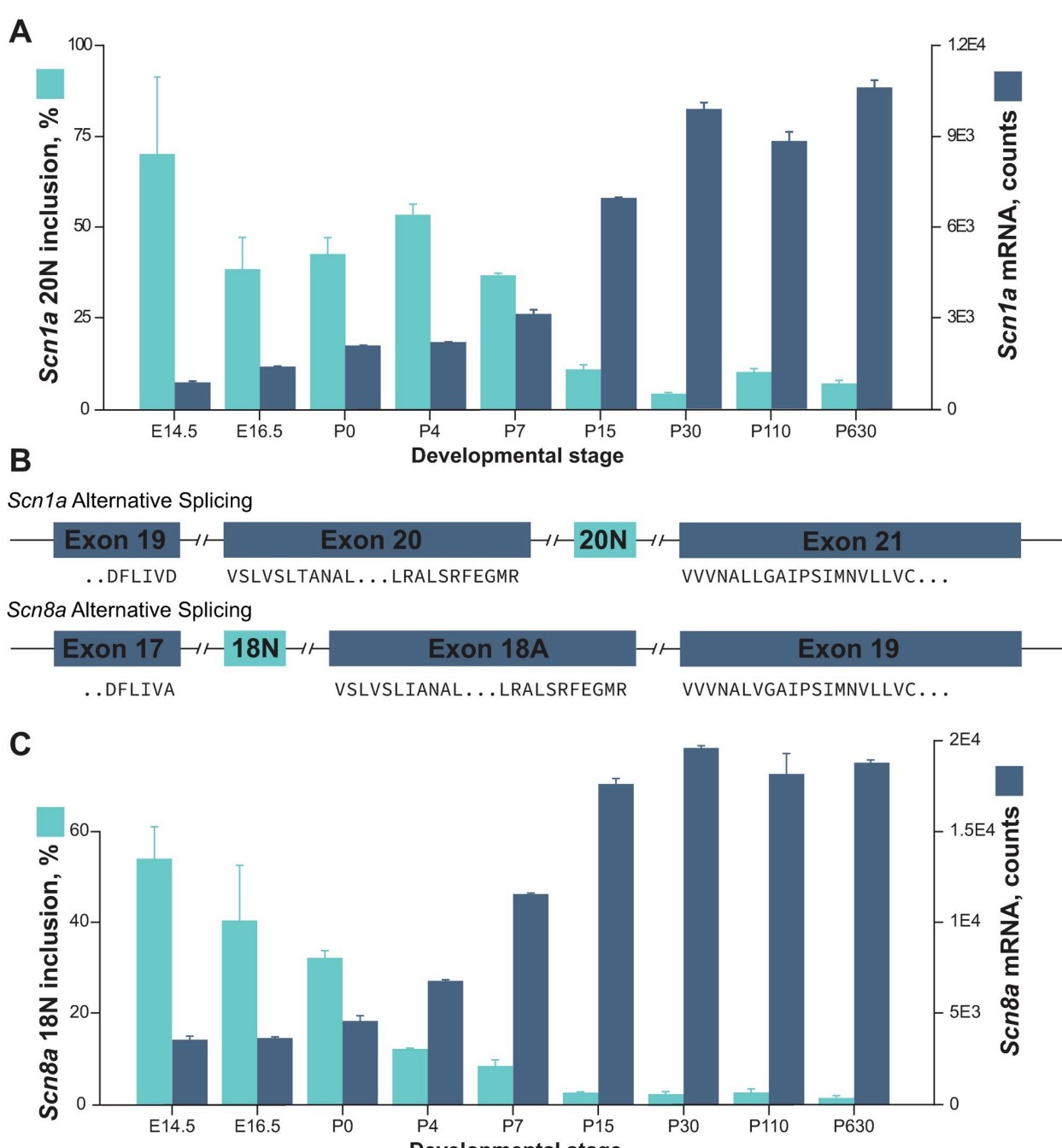

**Fig 5. Inverse relationship between poison exon usage and expression of multiple sodium channels during mouse brain development.** (A) *Scn1a* transcripts including exon 20N are highly expressed in the developing mouse brain and decrease dramatically after birth (aqua bars), with a corresponding developmental increase in *Scn1a* expression (blue bars). (B) The poison exon in *Scn8a* previously described by Plummer et al.[29]. *Scn1a* exon 20N and *Scn8a* exon 18N are 37.5% identical (57% in human), and the amino acid sequences shown at exon boundaries are identical. The amino acid sequences shown are fully identical between mouse and human for both genes. (C) *Scn8a* transcripts including exon 18N are highly expressed in the developing mouse brain and decrease dramatically after birth (aqua bars), with a corresponding developmental increase in *Scn8a* expression (blue bars).

*in vivo*. Introduction of this variant led to a reduction in brain *Scn1a* mRNA and Na$_v$1.1 protein levels, resulting in expression of DS-related phenotypes (Figs 2–3, Supplementary File 1). Inclusion of poison exon 20N in adult brain from *Scn1a* +/+ mice was ~1% and increased ~

fivefold in *+/KI* mice. Together with additional analyses and earlier work [29], our results suggest that poison exons in at least two neuronal sodium channels serve an important function in developmental regulation, suppressing expression of functional sodium channels until later stages of brain development. The mechanism is evolutionarily conserved and represents a previously unrecognized potential source of Mendelian disease.

Mutations in the *SCN1A* gene are the most common cause of DS, accounting for 80% of patients. While 20% of patients still do not have a definitive molecular diagnosis after exome sequencing, our previous results identified a number of variants in and around exon 20N in multiple patients with DS/DEE [11]. Our results provide compelling evidence for pathogenicity of the variant reported here and, by extension, additional variants that may enhance inclusion of exon 20N. Our *in vivo* finding that poison exon 20N inclusion leads to *Scn1a* loss of function explains how the phenotype of *Scn1a +/KI* mice mimics that of other DS models, since reduced *Scn1a* expression is the common feature underlying DS. *Scn1a* loss of function results in decreased expression of $Na_v1.1$, a voltage gated sodium channel responsible for promoting electrical excitability of neurons [33]. Loss-of-function *Scn1a* mutations would decrease neuronal activity. *Scn1a* is predominantly expressed in inhibitory GABAergic interneurons [7], so DS mutations decrease activity of the inhibitory circuitry and resulting disinhibition contributes to seizure generation [33].

In the model described here as well as other DS models on a C57BL/6J background [6, 7], seizure onset in heterozygous animals begins in the 3rd and 4th postnatal week, in and around the time the animals are weaned. This is consistent with both the normal onset of rodent $Na_v1.1$ expression in the second postnatal week as determined by mRNA or protein abundance [26, 34], and with our analyses of the time course of normal exon 20N inclusion, which reaches a nadir in the second postnatal week of life, and is inversely related to the abundance of normal *Scn1a* mRNA.

In humans, seizure onset in Dravet syndrome can be as early as 4–6 months of age [1, 35, 36], and the time course of transcriptional events is analogous to that in the mouse, with *SCN1A* mRNA expression accumulating during and after infancy, and exon 20N inclusion declining from 60–80% during embryonic development to 5–10% by 6 months of age [37]. Similarly, the switch from inclusion to exclusion of poison exon 18N takes place around the second postnatal week in rodent *Scn8a* and 9–12 months of age in human *SCN8A* [31]. Taken together with previous work on cell- and cell region-type specificity of $Na_v1.1$ and $Na_v1.6$ expression [4–8], these observations suggest that poison exons in voltage-gated sodium channels represent a normal and precisely regulated mechanism to ensure that different channels are expressed at the correct time and place in the developing nervous system, and provide a mechanistic explanation for the time at which clinical features due to deficiency for *SCN1A* or *SCN8A* first become apparent.

The developmental mechanisms that regulate exon 20N inclusion in *Scn1a* are likely similar to what has been reported previously for exon 18N in *Scn8a*, in which ubiquitously expressed splicing factors such as SRFS1 and SRFS2 as well as cell-specific factors such as RbFox-1 bind to specific sites in the pre-mRNA alternative exon and/or flanking regions to modulate splicing [31, 38]. In the case of *Scn8a*, inclusion of poison exon 18N has been suggested as a mechanism to prevent expression of a functional $Na_v1.6$ channel in non-neuronal cells [31]. However, we found no evidence that poison exon 20N serves an analogous function for *Scn1a*, since in the *+/KI* mice, postnatal expression of 20N is only detectable in the brain. Thus, the biological rationale for regulation of exon 20N in *Scn1a* may be to regulate developmental and cell type-specific expression within neuronal tissues, with more conventional mechanisms, e.g. chromatin accessibility, modulating transcriptional initiation, used to control neuronal vs. non-neuronal expression. It is interesting to note that mutations in *Scn1a* have been proposed to

contribute to cardiac arrhythmias and sudden unexpected death in epilepsy (SUDEP) based on studies in mice, but the relative roles played by direct and indirect mechanisms are not yet clear [39, 40].

A better understanding of the molecular pathogenesis in DS is critical to developing disease-modifying therapies. With regard to the variant we studied and its effects on poison exon inclusion, there is a growing list of therapeutic strategies that target mechanisms related to alternative splicing. An RNA-based therapeutic triggering poison exon inclusion in *DHX9*, a gene involved in Ewing sarcoma, has been exploited to enhance the efficacy of chemotherapy in cancer patients [41]. Additionally, in Duchenne muscular dystrophy and spinal muscular atrophy (SMA), oligonucleotides that promote exon skipping and alternative exon inclusion, respectively, are becoming increasingly applied in clinical settings [42, 43]. The *Scn1a +/KI* mouse model we have developed based on patients with non-coding variants may provide support to assess poison exon 20N targeting therapeutics postnatally and other methods that may lead to an increased expression of Na$_v$1.1.

In summary, our data indicate that poison exon inclusion is a conserved mechanism to suppress gene expression that is induced by an intronic mutation in *SCN1A* leading to DS. These findings deepen our understanding of the molecular genetic mechanisms leading to DS and provide a new mouse model for studying the effects of a novel intronic mutation. Further, confirmation of the relevance of poison exon inclusion to a Mendelian disorder, coupled to the observation that multiple genes are regulated by this mechanism during development, suggests that variation affecting poison exons may be more broadly relevant to human disease.

## Materials and methods

### Ethics statement

All experimental protocols were approved by the Institutional Animal Care and Use Committee of the University of Alabama at Birmingham.

### Conservation assessment

**GERP analysis.**   The pre-mRNA diagram constructed using the R package *ggbio* [44, 45] with coordinates obtained from the UCSC Genome Table Browser [46]. Conservation was analyzed across the region using GERP (Genomic Evolutionary Rate Profiling) conservation scores for each position [47]. Positive GERP scores reflect a high level of conservation, while negative GERP scores reflect a neutral rate of substitution at the region.

**Cross-species conservation analysis.**   Multiple sequence alignments were retrieved from the Vertebrate Multiz Alignment & Conservation [48] (100 Species) track of the UCSC Genome Table Browser [46].

### Animals

Mice were on an isogenic C57BL/6J background. Mice were housed in a pathogen-free barrier facility on a 12-hour light/dark cycle with ad libitum access to water and food (NIH-31 Open Formula Diet, #7917, Harlan). Mice were genotyped by PCR from tail tissue collected at weaning and at death. Both male and female mice were used in experiments. Littermate siblings were used as controls in each experiment. Experiments were completed by blinded investigators. For postmortem analyses, mice were anesthetized by Fatal-plus (Vortech) and perfused with 0.9% saline. Tissues were then removed, weighed, and dissected for processing as described below.

## Targeted CRISPR *Scn1a +/KI* generation

The variant used in our study, NM_006920.6(SCN1A):c.3969+2451G>C was a *de novo* mutation with CADD and GERP scores of 16.51 and 5.43, respectively, present in proband 3 in Carvill et al., 2018 [11].

The CRISPR guide was generated using an online tool (http://crispr.mit.edu). A mouse carrying the NC_000068.7:g.66293870C>G (GRCm38.p6) mutation was generated at the University of Alabama at Birmingham (UAB) Transgenic & Genetically Engineered Models Core (TGEMs). The reagents used were from Integrated DNA Technologies (IDT), Inc., Coralville, Iowa: Alt-R S. p. Cas9 Nuclease 3NLS (Cat # 1074181), Alt-R CRISPR tracrRNA (Cat # 1072533), Alt-R CRISPR crRNA (sequence: 5'-TTGCTCCAACTTGGATGGGG-3'), single-stranded donor oligonucleotide (ssODN) (sequence: 5'-A*C*A*TAAGTCACAGTGCAAGGATTAAAGGTAGCAAAAG GGGTAATACAGTACCCATAATAAAGGGCTGAGGGGAGGAACCACCGCTCCACgCCAT CCAAGTTGGAGCAAGATTATCCTATATAAAATAG*A*A*A-3').

The Cas9 protein-crRNA-tracrRNA RNPs were assembled according to the manufacturer supplied instructions. Injections, manipulations of C57BL/6 embryos and subsequent maintenance of mice lines were carried out at the TGEMs facility using protocols and methods compliant with the University of Alabama at Birmingham Institutional Animal Care and Use Committee and the Guide for the Care and Use of Laboratory Animals published by the National Institutes of Health.

About 150 blastocysts were injected. Of these 28 F0 pups survived up to weaning and were genotyped for the engineered mutation. Six of these carried the mutation. Three of these were chosen for further crosses as these had the strongest peak for the changed base and therefore likely to have less mosaicism.

## Genotyping

DNA from weaned mouse tails was prepared using the UAB Transgenic & Genetically Engineered Models Core Facility protocol: tail clips were incubated at 65°C for 12 hours in lysis buffer (10mM Tris, 75mM NaCl, 25mM EDTA, 1% SDS, 0.5mg/ml Proteinase K) with intermittent shaking. DNA was extracted using Phenol:Chloroform, precipitated with ethanol, dried and resuspended in TE buffer. DNA was quantified using the Thermofisher Qubit dsDNA HS Assay Kit (Q32854).

PCR was performed on tail clip DNA using NEB OneTaq DNA polymerase (M0480L) or NEB Phusion DNA Polymerase (M0530L) on an Applied Biosystems GeneAmp PCR System 9700. The following forward and reverse oligos were used to amplify and Sanger sequence the 327-bp target region: 5'-TGTCCCTACTGTGGTGCAAT-3' and 5'-CCCAAGCTGGGAAAATCGTAA-3'. Sanger sequencing was performed at Molecular Cloning Laboratories (MCLAB).

## Mouse tissue RNA extraction

RNA was extracted from frozen *Scn1a +/+* and *Scn1a +/KI* mouse tissues using the Norgen Biotek Corp Animal Tissue RNA purification kit (Cat # 25700) using the manufacturer supplied protocol. Genomic DNA contamination was reduced/eliminated using Thermofisher TURBO DNA-free Kit (AM1907). RNA was quantified using Thermofisher Qubit RNA BR Assay Kit (Q10211).

## cDNA synthesis

Random primed and oligo-dT primed cDNA was synthesized from total RNA using Thermofisher SuperScript IV First-Strand Synthesis System (18091050) using the manufacturer supplied protocol. The Random primed cDNA was used for qPCR analyses.

## qRT-PCR

Quantitative RT-PCR was performed on an Applied Biosystems QuantStudio 6 Flex using Thermofisher PowerUp SYBR™ Green Master Mix (A25742). The oligos were ordered from IDT Technologies after using their PrimerQuest oligo design tool. qPCR oligos were designed to amplify 3 amplicons. Amplicon 1 was amplified using the oligos 5'-CCCTAAGAGCCT-TATCACGATTT-3' and 5'-TAACAGGGCATTCACAACCA-3'. These were intron spaning oligos on exons 20 and 21 respectively. Amplicon 1 amplifies 2 products: a 56-bp product in trasncripts without exon 20N or a 120bp product in transcripts with exon 20N. Amplicon 2 was amplified using the oligos 5'-CGATTTGAAGGGATGAGGGATAA-3' and 5' GCATT-CACAACCACCCATAATAAA-3'. These exon-exon junction oligos span exons 20-20N and exons 20N-21, respectively. Amplicon 2 produces a 96-bp product only in the presence of 20N containing transcripts. Amplicon 3 was amplified using the oligos 5'-CTGGTGTTGGCTA-GACTTCTT-3' and 5'-GCTCTTAGTGTCCTTAGGGATTT-3'. These oligos are located on exons 19 and 20, respectively and amplify a 111-bp product.

Oligos for the housekeeping gene *Tata box binding protein* (*Tbp*) were ordered from IDT Technologies (Mm.PT.39a.22214839).

Selected amplified DNA from the qPCR was analyzed on the Agilent 2100 Bioanalyzer using the Agilent DNA 1000 Kit (5067–1504).

## RNA-seq

RNA-seq libraries were prepared from whole brain total RNA from 4 *Scn1a +/+* and 4 *Scn1a +/KI* mice using the Lexogen QuantSeq 3' mRNA-Seq Library Prep Kit FWD for Illumina. Briefly, first strand cDNA was synthesized from total RNA using oligo dT oligo containing an Illumina Read 2 linker. Then, the template RNA was digested and 2nd strand synthesized using random primer containing Illumina Read 1 linker and Unique Molecular Identifiers (UMIs). The libraries were then amplified using Illumina index containing oligos. Single-end sequencing was done on an Illumina NextSeq. The reads generated were deduplicated using the UMIs. The RNA-seq reads were trimmed using Trim Galore (https://github.com/FelixKrueger/TrimGalore). The trimmed reads were aligned to the mouse genome build GRCm38.p6 using STAR aligner [49]. HTseq was used to generate counts for genes from the alignments [50]. *Scn1a* and *Gapdh* counts were extracted from the normalized table of counts in the R package DESeq2 [51]. RNA-seq data is available at GEO GSE153461.

## Western blot

Left frontal cortex was sub-dissected and was flash frozen on dry ice, and then homogenized in 100 ul of homogenization buffer containing 50 mM Tris, pH 7.5, 150 mM NaCl, 5 mM EDTA, 1% TritonX-100, 0.1% sodium deoxycholate, 1:100 Halt Protease Inhibitor (Halt, Thermo-Fisher, 78438) and Phosphatase inhibitor 3 (Sigma Aldrich, P0044). The homogenate was then centrifuged at 5,000 speed for 10 minutes. The supernatant (S1) was transferred into new Eppendorf tube after the centrifugation. The supernatant (S1) was centrifuged again at the same conditions and the resulting supernatant (S2) was used to determine protein concentration using Bradford protein assay (Thermo Scientific, Pierce, Coomassie Plus (Bradford) Protein Assay, PI23238). Protein samples were prepared with 4x LDS (Life Technologies, NuPAGE LDS Sample Buffer (4X), NP0007) and 10x reducing agent (ThermoFisher, 10X Bolt Sample Reducing Agent, B0009), heated for 10 min at 70°C, then 10 μg were loaded and separated on 4–12% NuPage acrylamide gels (ThermoFisher, NuPAGE 10% Bis-Tris Midi Protein Gels, 26-well WG1203BOX) with NuPage MOPS running buffer for 1.5 hour at constant 150 V. Next, proteins were transferred to Immunobilon-FL PVDF membranes (Fisher, Millipore,

SLGV033RS) using NuPage transfer buffer transfer system overnight at constant 30 V. The membrane was blocked in 50% Li-Cor Odyssey buffer (Li-Cor, 927–40000) in tris-buffered saline with 0.1% Tween (TBS-T) blocking buffer for 1 hour at room temperature and incubated with the appropriate primary antibody. The specific primary antibodies were diluted in 50% Odyssey blocking buffer in TBS-T as follows: anti-SCN1A ($Na_v$1.1) (Alomone Labs, ASC-001, 1:1,000, overnight), anti-SCN1a ($Na_v$1.1) (Antibodies Incorporated, 75–023, 1:1,000, overnight), anti-GAPDH (Millipore, MAB374, 1:5,000, 1hour), anti-Actin (Cell Signaling, 4967S, 1:1,000, 1hr). After primary antibody treatment, membranes were washed three times in TBS-T followed by incubation for 1 hour with Alexa Fluor 700- or 800- conjugated goat antibodies specific for mouse immunoglobulin G (1:20,000, Li-COR). Membranes were then washed three times in TBS-T, followed by a single wash in TBS, imaged on the LI-COR Odyssey fluorescence imaging system, and quantified using Li-CPR Image Studio.

## Behavioral assessment

For all behavioral tests, experiments were carried out during light cycle at least one hour after the lights came on. All mice were transferred to testing room for acclimation at least one hour prior to experiments. Testing apparatuses were cleaned by 75% ethanol between experiments and disinfected by 2% chlorohexidine after experiments were finished each day. All mice were tested in all the behavioral tests in the same order. Investigators were blind to the genotype of individual mouse at the time of experiment.

**Open field.** Each mouse was placed into the corner of an open field apparatus (Med Associates) and allowed to walk freely for 10 minutes. Total and minute by minute ambulatory distance, jumps, stereotypic behavior counts, and percent time in center of each mouse were determined using the manufacturer's software.

**Elevated plus maze.** Elevated Plus Maze (Med Associates) has two open arms and two closed arms. Mice were placed in the hub of the maze and allowed to explore for five minutes. The time in each arm, as well as entrances to each arm, explorations, and head dips over the edge of the maze, were monitored by video tracking software (Med Associates).

**Y maze.** The Y maze apparatus consisted of three 15-inch long, 3.5-inch wide and 5-inch high arms made of white opaque plexiglass placed on a table. Each mouse was placed into the hub and allowed to freely explore for 6 minutes, with video recording. An entry was defined as the center of mouse body extending 2 inches into an arm, using tracking software (CleverSys). The chronological order of entries into respective arms was determined. Each time the mouse entered all three arms successively (e.g. A-B-C or A-C-B) was considered a set. Percent alternation was calculated by dividing the number of sets by the total number of entries minus two (since the first two entries cannot meet criteria for a set). Mice with 12 or fewer total entries were excluded from spontaneous alternation calculations due to insufficient sample size.

**Tube test for social dominance.** The tube test for social dominance was conducted as previously described [23]. Mice of the same sex, but opposite genotype, were released into opposite ends of a clear plastic tube and allowed to freely interact. Under these conditions, one mouse will force the other out of the tube. The first mouse with two feet out of the tube was considered to have lost the match. Each mouse was paired with three different opponents of the opposite genotype, and the winning percentage was calculated for each mouse by dividing the number of wins by the total number of matches.

**Three-chamber sociability test.** The three-chamber sociability test was conducted as previously described [24]. Mice were allowed to freely explore a three-chambered testing apparatus for 10 min prior the introduction of wire cages containing a novel mouse (adult sex-

matched C57Bl/6J) or a novel object (Lego block). Investigation of the novel mouse and object was then monitored for 10 minutes using video tracking software (CleverSys).

### *Scn1a* and *Scn8a* in developing mouse brain

Publicly available RNA-seq data (SRA Accession # SRP055008) [32] was used to examine the expression of *Scn1a* and *Scn8a* in the developing mouse cortex. The RNA-seq reads were trimmed using Trim Galore (https://github.com/FelixKrueger/TrimGalore). The trimmed reads were aligned to the mouse genome build GRCm38.p6 using STAR aligner. HTSeq was used to generate counts for genes from the alignments. *Scn1a* and *Scn8a* counts were extracted from the normalized table of counts in R package DESeq2. To calculate the proportion of 20N containing transcripts, samtools depth was used to extract coverage across each base of each exon of *Scn1a* and *Scn8a* [52]. The average read count per base was calculated by dividing the total read count by the size in bp of the exons. The averages for the 20N exon and all the exons were calculated separately. After normalizing to the number of mapped reads in each sample, percent poison exon usage was calculated using the formula: (average depth of coverage of the poison exon) / (average depth of coverage of all exons) *100.

### Statistics

Levels of Na$_v$1.1 and GAPDH proteins were analyzed by Student's *t*-test. Behavioral tests were analyzed by Student's t-test or two-way RM-ANOVA specified in the figures dependent on the outcome measure. The survival data were analyzed by Kaplan-Meier statistics and post-hoc Log-rank (Mantel-Cox) test.

Two-tailed *p*-values were calculated for all analyses, and the cut-off for statistical significance was set at 0.05. GraphPad Prism 7 was used for all analyses. Data are presented as mean ± SEM (Standard Error of the Mean). Significance denoted as *$p < 0.05$, **$p < 0.01$, ***$p < 0.001$, ****$p < 0001$.

### Supporting information

**S1 Fig. MultiZ (UCSC Genome Browser) alignment surrounding *SCN1A* exon 20N.** (EPS)

**S2 Fig. Full length Western blots of protein levels in *Scn1a* +/KI and *Scn1a* +/+ mouse brains.** (A) Brain (frontal lobe) Scn1a protein levels in *Scn1a* +/KI vs. *Scn1a* +/+ mice using rabbit anti-Na$_v$1.1 (Scn1a) antibody from Alomone Labs. (B) GAPDH using anti-GAPDH antibody from Millipore was used as loading control. (C) Scn1a protein levels using anti-Na$_v$1.1 (SCN1A) UC-Davis antibody. (D) Actin using anti-Actin antibodies from Cell Signaling was used as loading control. (E) RNA-seq counts of *Gapdh* mRNA from DEseq2 analysis in whole brains of *Scn1a* +/KI and *Scn1a* +/+ mice (n = 4–4, 11.64 ± 2.90 months, Student's unpaired t-test, *p* = 0.67). (EPS)

**S3 Fig. *Scn1a* +/KI mice have no behavioral changes detected in the elevated plus and Y mazes.** (A) Time spent in open arms of the elevated plus maze (*n* = 6–8, 13.81 ± 0.47 months, Student's unpaired t-test, *p* = 0.1475). (B) Time spent in closed arms of the elevated plus maze (*n* = 6–8, 13.81 ± 0.47 months, Student's unpaired t-test, *p* = 0.0958). (C) Total arm entries of the elevated plus maze (*n* = 6–8, 13.81 ± 0.47 months, Student's unpaired t-test, *p* = 0.1577). (D) Correct alternations in the Y maze (*n* = 6–8, 13.81 ± 0.47 months, Student's unpaired t-test, *p* = 0.5888). (E) Total distance travelled during 5 min in the Y maze (n = 6–8, 13.81 ± 0.47

months, Student's unpaired t-test, $p = 0.1242$). All data are expressed as mean ± SEM.
(EPS)

**S4 Fig. *Scn1a +/KI* mice have no social behavior deficits detected.** (A) *Scn1a +/KI* mice win equally to the littermate *Scn1a +/+* mice in the social dominance tube test ($n = 6$–8, 13.81 ± 0.47 months, Student's unpaired t-test, $p > 0.9999$). (B) During habituation, mice of both genotypes had no preference to the side (top or bottom) of the three-chamber box ($n = 6$–8, 13.81 ± 0.47 months, two-way RM-ANOVA, interaction $p = 0.1484$, main effect of side $p = 0.04828$, main effect of genotype $p = 0.4806$). (C) During testing, mice of both genotypes had no preference to a Lego block or a stranger mouse (S) as measured by time spent in a specific chamber containing a Lego block or a stranger mouse ($n = 6$–8, 13.81 ± 0.47 months, two-way RM-ANOVA, interaction $p = 0.08994$, main effect of stranger mouse $p = 0.1339$, main effect of genotype $p = 0.2311$). (D) *Scn1a +/KI* mice were not significantly different from *Scn1a+/+* litter-mate controls in time spent around a cup containing stranger mouse compared to time spent around a cup containing a Lego object (n = 6–8, 13.81 ± 0.47 months, two-way RM-ANOVA, interaction $p = 0.1969$, main effect of stranger mouse $^*p = 0.0167$, main effect of genotype $p = 0.0867$). All data are expressed as mean ± SEM.
(EPS)

**S1 Movie. Seizure activity in *Scn1a +/KI* animal.**
(MP4)

**S1 Table. Source data.**
(XLSX)

## Acknowledgments

We thank Brian Roberts and Sarah Strange for preparing the Lexogen 3' RNA-seq libraries. We thank Rachael Vollmer for help maintaining the mouse colony and Shreya Kashyap for technical assistance using tube test behavioral analysis. This work was supported by the Evelyn F. McKnight Brain Institute.

## Author Contributions

**Conceptualization:** Gopal Battu, Gregory M. Cooper, Erik D. Roberson, Gregory S. Barsh.

**Data curation:** Stephanie A. Felker.

**Formal analysis:** Yuliya Voskobiynyk, Stephanie A. Felker, J. Nicholas Cochran.

**Funding acquisition:** Gregory M. Cooper, Erik D. Roberson, Gregory S. Barsh.

**Investigation:** Yuliya Voskobiynyk, Gopal Battu, Stephanie A. Felker, J. Nicholas Cochran, Laura J. Lambert, Erik D. Roberson, Gregory S. Barsh.

**Methodology:** Yuliya Voskobiynyk, Gopal Battu, J. Nicholas Cochran, Megan P. Newton.

**Project administration:** Laura J. Lambert, Robert A. Kesterson, Richard M. Myers, Erik D. Roberson, Gregory S. Barsh.

**Resources:** Gopal Battu, Gregory M. Cooper.

**Supervision:** J. Nicholas Cochran, Robert A. Kesterson, Gregory M. Cooper, Erik D. Roberson, Gregory S. Barsh.

**Visualization:** Yuliya Voskobiynyk, Stephanie A. Felker.

**Writing – original draft:** Yuliya Voskobiynyk.

**Writing – review & editing:** Yuliya Voskobiynyk, Gopal Battu, J. Nicholas Cochran, Gregory M. Cooper, Erik D. Roberson, Gregory S. Barsh.

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
