## [Decision Letter · Decision Letter 0]

30 Jul 2020

Dear Dr Barsh,

Thank you very much for submitting your Research Article entitled 'Aberrant regulation of a poison exon caused by a non-coding variant in a mouse model of Scn1a-associated epileptic encephalopathy' to PLOS Genetics. Your manuscript was fully evaluated at the editorial level and by independent peer reviewers. The reviewers appreciated the attention to an important topic but identified some aspects of the manuscript that should be improved.

We therefore ask you to modify the manuscript according to the review recommendations before we can consider your manuscript for acceptance. Your revisions should address the specific points made by each reviewer.

The reviewers have suggeested minor revisions.

Yours sincerely,

Joseph Shieh, M.D. Ph.D.

Guest Editor

PLOS Genetics

Gregory P. Copenhaver

Editor-in-Chief

PLOS Genetics

Reviewer's Responses to Questions

**Comments to the Authors:**

Reviewer #1: Voskobiynik et al present their findings in support of a "poison exon" model for some patients with Dravet syndrome, and show that an intron 20 mutation found in patients recapitulates a Dravet syndrome-like phenotype in a CRISPR/Cas9 knockin mouse model. Overall, their data are compelling that the intronic mutation c.3969+2451G>C leads essentially to a haploinsufficiency state for SCN1A. They argue, though do not prove, that this is due to nonsense-mediated decay. Building on previous work with a poison exon in SCN8A, they also show that the poison exon is expressed during fetal development, and only becomes a minor species postnatally.

It would be helpful if the authors might briefly expand on the time points where exon 20N becomes a minor species (P15), and when the Dravet phenotype emerges. Is the increased death in SCN1A+/KI mice seen only around P15, but not at P7, for example?

There is also some literature to suggest an association of SCN1A mutations with SUDEP and fatal cardiac arrhythmias, and possibly liver toxicity. Perhaps, although the authors find lower SCN1A expression in heart and liver, these lower expressions should not be dismissed in the pathophysiology of DS. The 20N exon mRNA is undetectable in several of the non-brain tissues. Does this suggest that the poison exon does not explain reduced SCN1A expression in these tissues compared to brain?

The discussion of targeting exon 20N with therapeutics to treat DS is intriguing, though the mechanism of exon 20 formation likely needs to be clarified. This work would certainly be strengthened by mechanistic data regarding the incorporation of exon 20 in normal development and the increased incorporation in patients with the c.3969+2451G>C mutation. However, I do not think this additional mechanistic data is essential for the authors to argue their case in this study.

A minor point: SMA pathophysiology is not explained correctly in the discussion: SMA is not caused by haploinsufficiency of SMN2, but by homozygous loss of SMN1, with copy number of SMN2 influencing severity and onset of disease. It is true that nusinersen, an antisense oligonucleotide, is used to treat SMA by modifying splicing (incorporation of exon 7). This should be corrected in the manuscript, along with a few grammatical errors. Overall, the manuscript is clear and well-written.

Reviewer #2: oskobiynyk and collaborators report on the functional effects of a "poison exon" variant in SCN1A, a recently identified genetic mechanism in Dravet Syndrome. The authors report on the effect on a mouse model, replicating the phenotype and provide a conceptual framework on how a deep intronic variant affects poison exon inclusion and gene regulation. Their work is very detailed and provides a good rationale for this new genetic mechanism. I have the following comments and questions.

Seizure phenotype of SCN1A+/KI mice

The authors provide relatively little information on the seizure phenotypes of the SCN1A+/KI mice and do not provide some of the detailed EEG analysis that was used in other mouse models for SCN1A. Could the authors provide some more quantitative information on seizures in the SCN1A+/KI mice to make the case that their model aligns with prior SCN1A mouse models? Given the overall complexity of the manuscript, I would not require the authors to generate additional data, but simply provide existing data in a more comparative way.

Reference to clinical features of index patients

The authors refer to their index patient with the c.3969+2451G>C variant, but the reader is required to return to the prior publication by Carvill and collaborators to obtain more information. Could the authors provide a brief synopsis of the patient's clinical features and genetic features (de novo status of the c.3969+2451G>C variant)?

Survival of SCN1A+/KI mice

The authors provide a sufficient description on how SCN1A+/KI mice replicate prior Dravet Syndrome mouse models. However, following the description alone is relative difficult at times. Would it be possible for the authors to add information on "historic comparisons" for other DS models to Figure 3 (or some parts of Figure 3), for example by adding another survival curve in Figure A to compare SCN1A+/KI mice to prior DS models?

Ratio of inclusion of poison exon

In lines 171-174, the authors conclude that nearly all transcripts from the allele contain exon 20N compared to only 1% in WT mice. Could the authors expand on how they arrive at this number? It becomes obvious from comparing the reduction in overall SCN1A expression, but is this the only conclusion? Could the exon 20N mutation have triggered other transcriptional events that reduce SCN1A expression? Also, could the authors provide hypothesis how the increased inclusion is achieved? Is there anything known about regulatory elements in exon 20N that may have triggered increased/decreased binding of RNA-binding proteins that lead to the inclusion?

Developmental trajectory of poison exon inclusion

The authors should be complemented for their bioinformatic analysis of exon 20N inclusion in SCN1A and SCN8A. This data shows that 'poison exon inclusion' is not a pathological mechanism in itself, but that the disease mechanism in SCN1A and SCN8A is 'persistent poison exon inclusion', a defect of turning off a likely physiological developmental mechanism. The authors can consider whether they want to point out this aspect (optional).

Mouse vs. human development

Could the authors provide references on how mouse development corresponds to human development with respect to SCN1A expression. It may be implied from their paper that physiological poison exon inclusion ends at an estimated 3-6 months age in human when clinical features of DS become apparent, but a clinical reader may not be able to infer the human age at mouse age P15.

Reviewer #3: The authors developed another mouse model for Dravet syndrome (DS) based on knock-in of a variant in poison exon 20N in the gene SCN1A which they described previously in a DS patient. They nicely show that the mechanism is evolutionary conserved and that overexpression of the allele with a poison exon is triggered by the mutation and leads to a decrease of Nav1.1 channels of about 50%, comparable to other DS models. They further show that there is developmental regulation of Nav1.1 including poison exon 20N which contributes to the relatively late upregulation of Nav1.1 in development. A similar mechanism is confirmed for Nav1.6 which had been described previously. The results are clearly presented and the manuscript is well written

Critique

Major points

1. The manuscript stays on a descriptive level and the potential gene regulatory mechanisms are neither examined nor discussed. The authors describe that Nav1.1 expression in other than neuronal tissues is low throughout development, suggesting that the SCN1A promoter is responsible for tissue-specific expression of the gene and that the poison exon does not contribute to suppression of Nav1.1 expression in other tissues. The authors do not discuss this issue. I am not an expert in gene regulation but the authors mention other diseases like cancer in which poison exons play a role. Thus it may be that mechanisms have been examined as to why poison exons are expressed at different levels under various conditions, such as a disease-causing mutation or developmental regulation. At the very least, the authors should mention known mechanisms of gene regulation by poison exons and they should discuss different roles of promoter- and poison-exon-mediated mechanisms.

Minor points

2. Appropriate original literature should be cited on DS/SCN1A, not only reviews (refs 5,6). Claes et al. AJHG 2001, Yu et al. Nat Neurosci 2006, Ogiwara et al. J Neurosci 2007, Hedrich et al. J Neurosci 2014 are key papers with regard to genetics and pathophysiology of SCN1A defects including widespread interneuron and network dysfunction which should be cited right at the beginning.

3. The authors should mention in the results section that premature death in DS mouse models largely depends on genetic background and they should report the background they used not only in the methods section.

4. Fig. 1D: The linkers between segments S4 and S5 have a similar length in all four domains. The figure suggests that it is longer in domain 3, which is not the case. Please adapt the figure accordingly.

**Have all data underlying the figures and results presented in the manuscript been provided?**

Reviewer #1: Yes

Reviewer #2: Yes

Reviewer #3: Yes

PLOS authors have the option to publish the peer review history of their article (what does this mean?). If published, this will include your full peer review and any attached files.

Reviewer #1: No

Reviewer #2: No

Reviewer #3: No

---

## [Decision Letter · Decision Letter 1]

14 Oct 2020

Dear Dr Barsh,

We are pleased to inform you that your manuscript entitled "Aberrant regulation of a poison exon caused by a non-coding variant in a mouse model of Scn1a-associated epileptic encephalopathy" has been editorially accepted for publication in PLOS Genetics. Congratulations!

One of the reviewers suggested that you add a reference to the recently published Han et al., Sci. Transl. Med. 12, eaaz6100 (2020) paper. We agree that this would be a sensible addition and encourage you to do so as you prepare the final draft for the production team (the editorial team will not need to re-evaluate).

Yours sincerely,

Joseph Shieh, M.D. Ph.D.

Guest Editor

PLOS Genetics

Gregory P. Copenhaver

Editor-in-Chief

PLOS Genetics

Comments from the reviewers (if applicable):

Reviewer's Responses to Questions

**Comments to the Authors:**

Reviewer #1: The authors have addressed the reviewers' concerns, and I commend them on a well-developed demonstration of a poison exon mechanism for Dravet syndrome. The additions to the manuscript and figures, including additional explanations of the seizure phenotype and potential mechanisms of exon 20 inclusion, are much appreciated and strengthen this work. Well done!

Reviewer #2: The authors have responded to all my earlier questions and I have no further concerns.

Reviewer #3: The authors have generally addressed the comments of the reviewers satisfactorily. Unfortunately, the changes by the authors were NOT indicated in red, as they wrote at the beginning of their reply.

Since the first review, a manuscript was published that uses TANGO to suppress exon 20N, so what the authors propose in the discussion as a therapeutic strategy has been already done. The authors have to cite this paper in their revised version: Han et al., Sci. Transl. Med. 12, eaaz6100 (2020).

**Have all data underlying the figures and results presented in the manuscript been provided?**

Reviewer #1: Yes

Reviewer #2: Yes

Reviewer #3: Yes

PLOS authors have the option to publish the peer review history of their article (what does this mean?). If published, this will include your full peer review and any attached files.

Reviewer #1: No

Reviewer #2: No

Reviewer #3: **Yes: **Holger Lerche

**Data Deposition**

http://datadryad.org/submit?journalID=pgenetics&manu=PGENETICS-D-20-01021R1

**Press Queries**

---

## [Editor Report · Acceptance letter]

1 Dec 2020

PGENETICS-D-20-01021R1 

Aberrant regulation of a poison exon caused by a non-coding variant in a mouse model of Scn1a-associated epileptic encephalopathy 

Dear Dr Barsh, 

We are pleased to inform you that your manuscript entitled "Aberrant regulation of a poison exon caused by a non-coding variant in a mouse model of Scn1a-associated epileptic encephalopathy" has been formally accepted for publication in PLOS Genetics! Your manuscript is now with our production department and you will be notified of the publication date in due course.

With kind regards,

Nicola Davies

PLOS Genetics

On behalf of:
